# Critical Conditions for Wellbore Failure during $CO_2$-ECBM Considering Sorption Stress

Hecheng Xiao [1,2], Wenda Li [1,*], Zaiyong Wang [1,2], Shuai Yang [3] and Peng Tan [4,5]

1   Key Laboratory of In-Situ Property-Improving Mining of Ministry of Education,
    Taiyuan University of Technology, Taiyuan 030024, China
2   College of Mining Engineering, Taiyuan University of Technology, Taiyuan 030024, China
3   PetroChina Research Institute of Petroleum Exploration & Development, Beijing 100083, China
4   CNPC Engineering Technology R&D Company Limited, Beijing 102206, China
5   National Engineering Research Center for Oil & Gas Drilling and Completion Technology,
    Beijing 102206, China
*   Correspondence: liwenda@tyut.edu.cn

**Abstract:** Significant stress changes caused by sorption-induced swelling raise the coal wellbore failure potential, which directly impacts the safety and sustainability of $CO_2$ enhanced coalbed methane ($CO_2$-ECBM). Additionally, a mixture gas ($CO_2/N_2$) injection is recommended due to the sharp decline of permeability with pure $CO_2$ injection. In this study, incorporating the impacts of mixture gas adsorption and poroelastic effects, a semi-analytical model of coal wellbore stability during mixture gas injection is proposed. Model results indicate that the stress field is significantly influenced by the boundary condition and sorption effect. In addition, parametric studies are performed to determine the influence of adsorption parameters, mechanical properties, and gas composition on the stress distribution and then on the wellbore failure index. Furthermore, mixture gas injection with a large proportion of $CO_2$ or $N_2$ both cause wellbore instability. Significant compressive hoop stress and shear failure are caused by the mixture gas injection with a large proportion of $CO_2$. In contrast, the displacement of $CH_4$ with weakly adsorptive $N_2$ will result in less compressive and even tensile hoop stress, so shear or tensile failure may occur. Thus, mixture gas (including pure $CO_2/N_2$) injection must be controlled by coal wellbore failure, providing an accurate estimation of in-situ coal seams' $CO_2$ storage capacity from the perspective of wellbore stability.

**Keywords:** wellbore stability; $CO_2$-ECBM; mixture gas injection; sorption stress; poroelastic response

## 1. Introduction

Since 95–98% of the total gas can be stored as absorbed gas in the microporous structure [1], deep, unmineable coalbeds with enormous $CO_2$ storage capacity are considered to be an appealing alternative for carbon dioxide receptacles [2,3]. While large-scale carbon capture and storage (CCS) must evaluate all potential modes of failure, stress changes due to $CO_2$ injection need to be limited [4]. In addition, significant stress changes caused by sorption-induced swelling also raise the coal failure potential [5], which directly impacts the safety and sustainability of $CO_2$-ECBM.

Coal failure during coalbed methane production has received a lot of attention from researchers, whereas coal failure during $CO_2$-ECBM has received less attention. Shi and Durucan [6] initially estimated the effective stress change during depletion using an analogy between thermal contraction and gas desorption-induced matrix shrinkage, and experimental data demonstrate excellent agreement [7]. Liu and Harpalani [8] provided an experimental investigation of the horizontal stress variations, and discovered that in addition to the poroelastic effect seen in conventional reservoirs, sorption-induced shrinkage also alters in-situ stress and increases coal failure potential. Additionally, experimental studies revealed that gas with a higher adsorption capacity corresponds to a greater horizontal

stress loss in depletion [9,10]. This in turn suggests that $CO_2$ injection would result in a greater horizontal stress change. The reservoir-scale failure of $CO_2$-ECBM was explored by Lu and Connell [11], who noted that stress changes are complex phenomena connected to reservoir pressure, gas mix, and adsorption parameters. Above all, however, these models account for constant vertical stress and a uni-axial strain condition. Due to the non-uniform pore pressure distribution in the reservoir, a numerical method was devised to discover that the common uni-axial strain condition has significant deviations in the coal failure analysis [12]. In order to properly analyze coal failure during $CO_2$-ECBM, non-uniform pore pressure distribution, adsorption characteristics, and gas composition must all be taken into account.

Furthermore, shear failure in the reservoir formation is promoted by near-wellbore stress concentration [13]. In addition, local wellbore failure occurred before reservoir-scale coal failure [14]. So, a basis for the safe $CO_2$-ECBM or sequestration is wellbore stability rather than coal stability, and it can provide an upper limit for $CO_2$ storage capacity. Cui et al. [15] developed analytical solutions of the stress field under the uni-aixal strain, general stress, and plane strain conditions, shedding light on the stress distribution in a reservoir with an isotropic horizontal stress state. In the process of CBM depletion, Reisabadi et al. [16] examined the distinct coal failure index of the coal seams under various stress regimes. Masoudian et al. [17] simulated the stress change in coal with respect to the effect of the sorption-induced modification on mechanical behavior. Hu et al. [18] also developed a model for the cement-formation interface failure length during $CO_2$-ECBM, demonstrating the importance of the injection rate in determining the interface failure.

Furthermore, pure $CO_2$ injection causes more sorption stress and permeability decline than $N_2$, so injecting a $CO_2/N_2$ mixture gas is recommended. Li and Elsworth [19] discovered that a higher proportion of $CO_2$ in the injected $CO_2/N_2$ mixture results in lower shale gas recovery. Wen et al. [20] investigated the process of $CO_2$ gas replacing $CH_4$ using low-field NMR technology, and the effects of injection pressure and temperature were analyzed. Moreover, a micro pilot test of underground displacement using $N_2$ injection demonstrated a significant stimulation effect when compared to a conventional emission test [21]. However, various adsorptive gases cause various sorption stresses, which change the formation and wellbore stability to a different extent. To the best of the authors' knowledge, wellbore stability of coal seams with anisotropic stress state during mixture gas injection (including pure $CO_2/N_2$) has not been thoroughly studied.

In this study, a semi-analytical method was applied to investigate the wellbore failure index during reservoir-scale $CO_2$-ECBM assuming zero displacement at the outer boundary condition and plane-strain condition near the wellbore. Furthermore, steady reservoir pressure distribution is assumed to simply simulate the mixture gas (or pure $CO_2$) injection process. The analytical total stress field is derived by considering sorption-induced swelling and poroelastic effect, superposing the stress caused by in- situ stress, borehole pressure, reservoir pressure distribution, and gas adsorption. The critical borehole pressure and gas component (CBPGC) is calculated by combining the Mogi–Coulomb failure criterion with the total stress field. The CBPGC can serve as a benchmark for numerical simulation and quick assessment of the $CO_2$ storage capacity of in- situ coal seams.

## 2. Model Development

The coal is a porous medium and adsorbent in which the stress state changes as the reservoir pressure and adsorption amount vary during $CO_2$-ECBM. In addition, the initial in- situ stress state, including the reservoir pressure, is essential to determine the ultimate stress state and failure potential. In this study, a wellbore with a radius of $r_w$ covering a cylindrical domain with a radius of $r_b$ is considered.

### 2.1. Stress–strain Constitutive Equation

Gases such as $N_2$, $CH_4$, and $CO_2$ are absorbed primarily on the internal surface of microporous coal particles, causing significant matrix swelling. Using and extending Cui's

model [15], the volumetric strain associated with mixture gas sorption can be written as follows

$$\varepsilon_v = \frac{\varepsilon_L b_L p}{1 + b_L p},$$ (1)

where

$$b_L = \sum \frac{c_i}{p_{\varepsilon i}}, \varepsilon_L = \frac{1}{b_L} \sum \frac{\varepsilon_{Li} c_i}{p_{\varepsilon i}},$$ (2)

Additionally, $p_{\varepsilon i}$ and $\varepsilon_{Li}$ are the Langmuir-type swelling constants of gas $i$, which represent the maximum swelling capacity and the pore pressure at which the volumetric strain is equal to $0.5\varepsilon_{Li}$, respectively [22,23]. Additionally, Langmuir-type swelling constants differ from Langmuir-type adsorption constants: $V_{Li}$ and $P_{Li}$ [9]. Additionally, $c_i$ is the mole fraction of gas component $i$ in free gas mixture, $p$ is the reservoir (pore) pressure. Moreover, Cui et al. [15] established a relationship by using the experiment data: $\varepsilon_{Li} = \varepsilon_{gi} V_{Li}, p_{\varepsilon i} = P_{Li}$ with $\varepsilon_{gi}$ volumetric strain coefficient of gas $i$, which gave similar values to those of Liu et al. [9]. In addition, $\varepsilon_{V0} = \frac{\varepsilon_{LCH_4} p_0 / p_{\varepsilon CH_4}}{1 + p_0 / p_{\varepsilon CH_4}}$ is the initial volumetric strain caused by $CH_4$ adsorption in the initial state, and $p_0$ is the initial reservoir pressure.

The pressure variation during $CO_2$-ECBM has two impacts on effective horizontal stresses. In addition to the poroelastic (poromechanical) effect, the sorption-induced matrix shrinkage/swelling also modifies the effective horizontal stresses. Similar to the constitutive law of the non-isothermal poroelastic medium, the constitutive equations of coal seam considering sorption can read [9,15,22,23],

$$\sigma_{ij} = \frac{E}{1 + \nu}(\varepsilon_{ij} + \frac{\nu}{1 - 2\nu}\varepsilon_b \delta_{ij}) + \varsigma p \delta_{ij} + \frac{E}{3(1 - 2\nu)}\varepsilon_V \delta_{ij},$$ (3)

where E is Young's modulus of porous medium, $\nu$ is Poisson's ratio, $\varepsilon_b$ is the bulk volumetric strain, $\delta_{ij}$ is Kronecker's delta and $i$ or $j$ is the directional index $r, \theta$ and $z$ in cylindrical coordinates. Additionally, $\varsigma$ is the Biot constants, which are assumed to be one due to the large amount of cleats and fractures in the coal.

### 2.2. Analytical Model of Stress Distribution around the Wellbore

The total stresses around the borehole during the $CO_2$-ECBM can be split into the two categories below: (1) Mode 1: the stresses induced by the in-situ principal stresses and the bottom-hole pressure, which is significant near the wellbore and referred to the "near wellbore effect"; (2) Mode 2: additional stresses due to fluid flow and gas sorption. Then, the analytical solution of the total stress around the wellbore can be obtained by superposition.

#### 2.2.1. Simplifications and Assumptions

For the gas injection, the reservoir pressure, gas concentration, and stress will change transiently. While as the gas injection continues, the reservoir pressure profile approaches a steady state, which is considered here and allows us to obtain simple solutions and a perceptive understanding. Differing to Cui's assumptions [15], the anisotropic in-situ stress state and depleted reservoir is considered here. The assumptions conclude:

(1) The tectonic stress affects the coal seams and the initial in-situ stress is anisotropic;
(2) A steady reservoir pressure profile changing logarithmically from a constant borehole pressure at the wellbore ($r_w$) to a reservoir pressure $p_{res}$ at the outer boundary ($r_b$);
(3) Gas injection can occur after depletion, in which case the reservoir pressure is unequal to the initial value before depletion ($p_{res} \neq p_0$);
(4) A zero displacement at the outer boundary rather than each point (common uni-axial strain model) in the reservoir [15];
(5) A plane strain condition near the wellbore and a constant overburden $\sigma_{zz}$ far from the wellbore [14];
(6) At the final steady state, uniform gas composition of the injected gas is achieved.

### 2.2.2. Mode 1: Stresses Induced by the Initial In-Situ Stress and Borehole Pressure

According to the classical Kirsch solution, the stress distribution around the wellbore for a vertical wellbore is given as [14,24,25]

$$
\begin{aligned}
\sigma_{rr}^1 &= p_w \frac{r_w^2}{r^2} + \frac{\sigma_{h0}+\sigma_{H0}}{2}\left(1 - \frac{r_w^2}{r^2}\right) + \frac{\sigma_{h0}-\sigma_{H0}}{2}\left(1 - 4\frac{r_w^2}{r^2} + 3\frac{r_w^4}{r^4}\right)cos2\theta, \\
\sigma_{\theta\theta}^1 &= -p_w \frac{r_w^2}{r^2} + \frac{\sigma_{h0}+\sigma_{H0}}{2}\left(1 + \frac{r_w^2}{r^2}\right) - \frac{\sigma_{h0}-\sigma_{H0}}{2}\left(1 + 3\frac{r_w^4}{r^4}\right)cos2\theta, \\
\sigma_{zz}^1 &= \sigma_{v0} - 2\nu(\sigma_{h0} - \sigma_{H0})\frac{r_w^2}{r^2}cos2\theta, \\
\sigma_{r\theta}^1 &= -\frac{\sigma_{h0}-\sigma_{H0}}{2}\left(1 + 2\frac{r_w^2}{r^2} - 3\frac{r_w^4}{r^4}\right)sin2\theta, \\
\sigma_{rz}^1 &= \sigma_{\theta z}^1 = 0
\end{aligned}
\tag{4}
$$

where $p_w$ is the borehole pressure, $\theta$ is the angular position around the wellbore. $\sigma_{H0}$, $\sigma_{h0}$, $\sigma_{v0}$ are the initial maximum, minimum horizontal and vertical stresses, which are determined by stress measurement in the initial in-situ state before depletion/depressurization. Additionally, the superscript "1, 2" denotes the stresses induced by the Mode 1 and 2, respectively.

### 2.2.3. Mode 2: Stresses Induced by Poroelastic Response and Gas Adsorption

When mixture gas ($CO_2/N_2$) is gradually injected into the coal seams, the pore pressure eventually approaches a steady state, from which we can derive an analytical solution. Assuming that the wellbore has a radius $r_w$ covering a cylindrical domain with a radius $r_b$ and a constant pressure, $p_{res}$ is fixed before injection, then the incremental steady reservoir pressure distribution relative to the initial reservoir pressure can be approximated as [15,26]

$$
\Delta p = p_{res} - p_0 + \frac{p_w - p_i}{\ln(r_w/r_b)}\ln\frac{r}{r_b},
\tag{5}
$$

where $p_w$ is borehole pressure, $p_{res}$ is the uniform reservoir pressure before mixture gas injection in the coal seams. It should be noted that appropriate injection timing can continue to the time when coal fails in depletion, so $p_{res} \leq p_0$ can be assumed. The incremental volumetric strain due to displacement of $CH_4$ with $CO_2/N_2$ can be described by the extended Langmuir model as

$$
\Delta\varepsilon_V = \frac{\varepsilon_L b_L p}{1 + b_L p} - \varepsilon_{V0},
\tag{6}
$$

Then, the isotropic elastic constitutive law in the incremental form of Equation (3) in the cylindrical coordinates reads

$$
\begin{aligned}
\Delta\sigma_{rr} &= \frac{E}{1+\nu}\left(\varepsilon_{rr} + \frac{\nu}{1-2\nu}\varepsilon_b\right) + \Delta p + \frac{E}{3(1-2\nu)}\Delta\varepsilon_V, \\
\Delta\sigma_{\theta\theta} &= \frac{E}{1+\nu}\left(\varepsilon_{\theta\theta} + \frac{\nu}{1-2\nu}\varepsilon_b\right) + \Delta p + \frac{E}{3(1-2\nu)}\Delta\varepsilon_V, \\
\Delta\sigma_{zz} &= 0 = \frac{E}{1+\nu}\left(\varepsilon_{zz} + \frac{\nu}{1-2\nu}\varepsilon_b\right) + \Delta p + \frac{E}{3(1-2\nu)}\Delta\varepsilon_V,
\end{aligned}
\tag{7}
$$

With

$$
\varepsilon_{rr} = \frac{\partial u}{\partial r}, \varepsilon_{\theta\theta} = \frac{u}{r}, \varepsilon_{zz} = \frac{\partial u_z}{\partial z}, \varepsilon_b = \varepsilon_{rr} + \varepsilon_{\theta\theta} + \varepsilon_{zz},
\tag{8}
$$

The stress equilibrium equation in Mode 2 is given as

$$
\frac{\partial\sigma_{rr}^2}{\partial r} + \frac{\sigma_{rr}^2 - \sigma_{\theta\theta}^2}{r} = 0,
\tag{9}
$$

Combining Equations (7) and (8) with Equation (9), we can obtain the incremental radial displacement

$$
\frac{\partial}{\partial r}\left[\frac{1}{r}\frac{\partial(ru)}{\partial r}\right] = -\frac{(1+\nu)(1-2\nu)}{E}\frac{\partial\Delta p}{\partial r} - \frac{1+\nu}{3}\frac{\partial\Delta\varepsilon_V}{\partial r},
\tag{10}
$$

Then, integrating Equation (10) two times yields the solution

$$u = -\frac{(1+v)(1-2v)}{E}\frac{F_p}{r} - \frac{1+v}{3}\frac{F_\varepsilon}{r} + \frac{rC_1}{2} + \frac{C_2}{r},$$

(11)

Then, induced radial, hoop stress due to fluid flow and gas sorption in the incremental form of Mode 2 is derived as follows

$$\sigma_{rr}^2 = \frac{(1-2v)}{r^2}F_p + \frac{E}{3r^2}F_\varepsilon + \frac{EC_1}{2(1-v)} - \frac{EC_2}{(1+v)r^2},$$

(12)

$$\sigma_{\theta\theta}^2 = -\frac{(1-2v)}{r^2}F_p - \frac{E}{3r^2}F_\varepsilon + (1-2v)\Delta p + \frac{E}{3}\Delta\varepsilon_v + \frac{EC_1}{2(1-v)} + \frac{EC_2}{(1+v)r^2},$$

(13)

With

$$F_p = \int_{r_w}^r \Delta p r dr = \frac{r^2}{4}[2(p_i - p_0) + q(2ln(\frac{r}{r_b}) - 1)],$$

(14)

$$F_\varepsilon = \int_{r_w}^r \Delta\varepsilon_v r dr = \frac{\varepsilon_L}{b_L}[(b_L - \varepsilon_{v0}\frac{b_L}{\varepsilon_L})\frac{r^2}{2} - \frac{1}{q}r_b^2 e^{-[2(1+b_L p_{res})]/b_L q}Ei(2(\frac{1+b_L p_{res}}{b_L q} + \ln(\frac{r}{r_b})))],$$

(15)

where $Ei(x) = -\int_{-x}^\infty e^{-t}/tdt$, $q = (p_w - p_{res})/\ln(r_w/r_b)$. Additionally, $C_1, C_2$ can be solved by the imposed boundary conditions. In addition, the term $F_p, F_\varepsilon$ in Equations (14) and (15) denotes the poroelastic and sorption effect on the radial and hoop stresses in Equations (12) and (13), respectively. When a stress boundary condition $\sigma_{rr}^2(r = r_w) = 0$ at the borehole wall and zero displacement condition ($u_r(r = r_b) = 0$) at the outer boundary (ZDBC) is specified, we can obtain from Equations (11) and (12)

$$\begin{cases} C_1 = -\frac{2(1-v)}{[(1+v)r_w^2 + (1-v)r_b^2]}\left[\frac{(1+v)(1-2v)}{E}F_p(r_b) + \frac{(1+v)}{3}F_\varepsilon(r_b)\right] \\ C_2 = \frac{(1+v)r_w^2}{2(1-v)}C_1 \end{cases}$$

(16)

When a constant stress boundary condition ($\sigma_{rr}^2(r = r_w, r_b) = 0$) at the borehole wall and outer boundary (ZSBC) is specified, i.e., no lateral constrain at the outer boundary, from Equations (11) and (12) we can obtain

$$\begin{cases} C_1 = -\frac{2(1-v)}{[r_b^2 - r_w^2]E}\left[(1-2v)F_p(r_b) + \frac{E}{3}F_\varepsilon(r_b)\right] \\ C_2 = \frac{(1+v)r_w^2}{2(1-v)}C_1 \end{cases}$$

(17)

The incremental stress equations derived here are similar to those developed by Cui et al. [15] in absolute form of total stress. However, Cui's model cannot study the wellbore stability of the coal seams with anisotropic in-situ stress state. In addition, $CO_2$ is often injected into the depleted coal seams, i.e., the reservoir pressure before injection is not equal to the initial reservoir pressure ($p_0 \neq p_{res}$). Additionally, the present model can deal with the situation.

2.2.4. The Total Stresses around the Wellbore with Anisotropic In-Situ Stress

Ultimately, the complete solution of stress field around a vertical wellbore during the $CO_2$-ECBM process can be obtained by superposition

$$\begin{aligned} \sigma_{rr} &= \sigma_{rr}^1 + \sigma_{rr}^2, \\ \sigma_{\theta\theta} &= \sigma_{\theta\theta}^1 + \sigma_{\theta\theta}^2, \\ \sigma_{r\theta} &= \sigma_{r\theta}^1, \\ \sigma_{zz} &= \sigma_{zz}^1, \end{aligned}$$

(18)

Under the linear elastic theory, the maximum stress concentration and then failure index usually appears at the wellbore wall [27,28]. So stress distribution and the following

failure index can be studied at the borehole wall. Then, the total stresses at the wellbore wall can be given as

$$\sigma_{rr}^w = p_w, \tag{19}$$

$$
\begin{aligned}
\sigma_{\theta\theta}^w = \quad & \sigma_{h0} + \sigma_{H0} - 2(\sigma_{h0} - \sigma_{H0})cos2\theta - p_w \\
& + (1-2\nu)(p_w - p_0) + \frac{E}{3}\left(\frac{\varepsilon_L b_L p_w}{1+b_L p_w} - \frac{\varepsilon_{LCH_4}p_0/p_{\varepsilon CH_4}}{1+p_0/p_{\varepsilon CH_4}}\right) \\
& + \frac{EC_1}{2(1-\nu)} + \frac{EC_2}{(1+\nu)r^2},
\end{aligned}
\tag{20}
$$

$$\sigma_{zz}^w = \sigma_{\nu 0} - 2\nu(\sigma_{h0} - \sigma_{H0})cos2\theta, \tag{21}$$

$$\sigma_{r\theta}^w = 0, \tag{22}$$

### 2.3. Failure Criterion

Additionally, an appropriate shear failure criterion for wellbore stability should be applied for coal failure evaluation. The Mogi–Coulomb failure criterion, considering the effect of intermediate principal stress, can match well with field data [14,16,29]. In this study, the Mogi–Coulomb failure criterion is used to forecast coal failure during $CO_2$-ECBM [14].

$$F = \tau_{oct} - a - \left(b\frac{S_1 + S_3}{2} - p\right), \tag{23}$$

$$a = \frac{2\sqrt{2}C\cos\varphi}{3}, \tag{24}$$

$$b = \frac{2\sqrt{2}C\sin\varphi}{3}, \tag{25}$$

$$\tau_{oct} = \frac{1}{3}\sqrt{(S_2 - S_3)^2 + (S_2 - S_1)^2 + (S_1 - S_3)^2}, \tag{26}$$

where $C, \varphi$ are the rock cohesion and friction angle, and $S_1 = \max\{\sigma_{rr}, \sigma_{\theta\theta}, \sigma_{zz}\}$, $S_2 = \text{median}\{\sigma_{rr}, \sigma_{\theta\theta}, \sigma_{zz}\}$, $S_3 = \text{minimum}\{\sigma_{rr}, \sigma_{\theta\theta}, \sigma_{zz}\}$, respectively. It should be noted that the $\sigma_{rr}, \sigma_{\theta\theta}, \sigma_{zz}$ in Equations (19)–(21) are the principal stresses at the borehole, since shear stresses at the borehole are $\sigma_{r\theta} = \sigma_{rz} = \sigma_{\theta z} = 0$.

Then, the total stresses are substituted into the failure criterion to obtain the critical borehole pressure and gas component (CBPGC), defining the critical allowable $p_w$ with specific gas components $c_{CO_2}$ in terms of wellbore stability during $CO_2$-ECBM.

### 3. Induced Stress during $CO_2$-ECBM and Parametric Analysis

Differing to the conventional reservoir, except for the poroelastic response, the swelling during adsorptive gas injection causes tremendous stress change, which is not well studied during $CO_2$-ECBM. So the stress change considering the poroelastic and adsorptive effect during mixture gas ($CO_2/N_2$) injection must be carefully researched. Additionally, the influencing factors' effect on the induced stress is also studied. Table 1 presents the typical input data in this study.

**Table 1.** Input parameters for simulation [9,15,16].

| Parameters | Variables | Values |
|---|---|---|
| Poromechanical parameters | Young modulus E (MPa) | 4.35 |
| | Poisson ratio $\nu$ | 0.3 |
| | Cohesion C(MPa) | 6.44 |
| | Friction angle $\varphi$ | 45 |
| | Biot coefficient $\varsigma$ | 1 |
| Adsorption parameters | Langmuir-type constants $\varepsilon_L$ | N$_2$: 0.0074; CH$_4$: 0.0106; CO$_2$: 0.0389 |
| | Langmuir-type constants $P_\varepsilon$(MPa) | N$_2$: 24.71; CH$_4$: 6.02; CO$_2$: 4.31 |

**Table 1.** *Cont.*

| Parameters | Variables | Values |
|---|---|---|
| In-situ stress state | Vertical stress $\sigma_{v0}$, MPa | 20.68 |
| | Maximum horizontal stress $\sigma_{H0}$(MPa) | 16.55 |
| | Minimum horizontal stress $\sigma_{h0}$(MPa) | 14.48 |
| | Initial reservoir pressure $p_0$(MPa) | 9.5148 |
| Injection parameters | Reservoir pressure before injection $p_{res}$(MPa) | 3 |
| | Borehole pressure $p_w$(MPa) | >3 |
| | Pure $CO_2$ injection $c_{CO_2}$ | 1 |
| | Mixture gas injection $c_{CO_2}$ | 0.8, 0.2, 0.1, 0.05 |
| | Mixture gas injection $c_{N_2} = 1 - c_{CO_2}$ | 0.2, 0.8, 0.9, 0.95 |
| | Wellbore radius $r_w$(m) | 0.1 |
| | Studied cylindrical domain radius $r_b$(m) | 2000 |

### 3.1. Model Verification

This paper aims to use the superposition method to provide an analytical solution considering the anisotropic in-situ stress state in the coal seams. Figure 1 illustrates the great agreement between the model results for the induced effective stress change and radial displacement due to $CO_2$ injection and Cui's model [15], verifying the accuracy of the present model. The induced effective horizontal stress change in Figure 1a can be given as $\Delta\sigma^{eff} = (\sigma_{rr}^2 + \sigma_{\theta\theta}^2)/2 - \Delta p$ because the induced effective stress change is caused by fluid flow and sorption. Furthermore, the ZDBC (zero displacement at the outer boundary) condition restricts the lateral deformation and results in an order-of-magnitude difference in radial displacement (seen in Figure 1b) compared to the ZSBC (zero stress change at the outer boundary). As a result, a more significant stress change is induced during $CO_2$-ECBM.

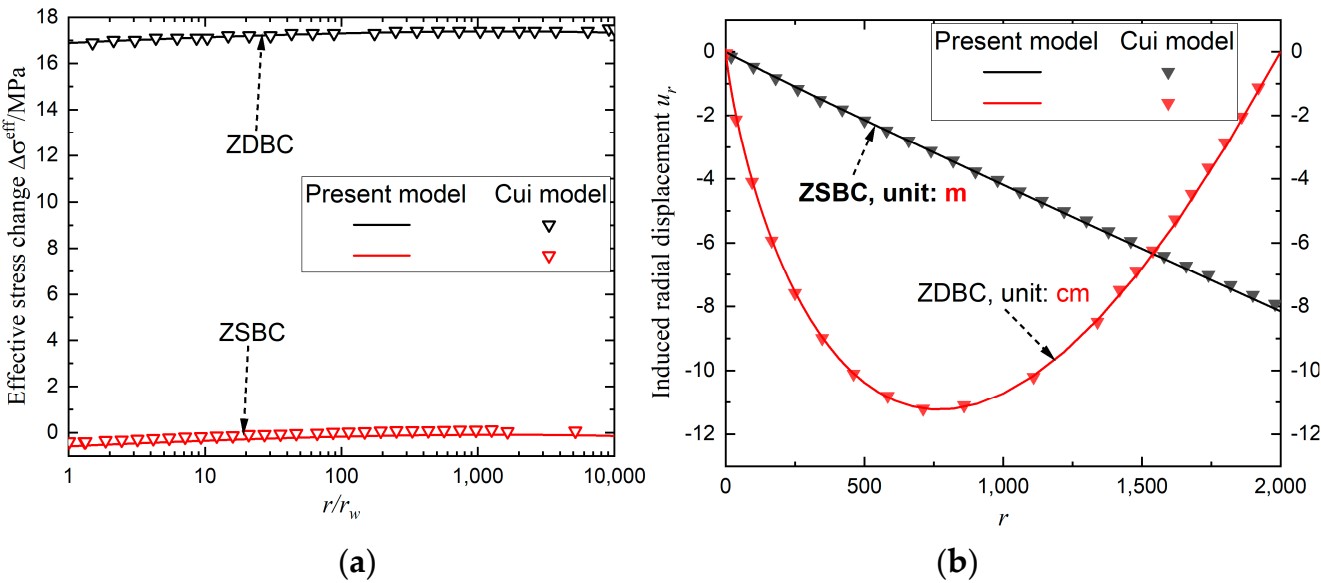

**Figure 1.** Profiles of (**a**) effective horizontal stress change and (**b**) induced radial displacement during pure $CO_2$ injection between the present model and results by Cui et al. [15]. ZDBC and ZSBC denote zero displacement and zero stress change at the outer boundary, respectively.

### 3.2. The Stress Caused by Poroelastic Response and Gas Adsorption

In contrast to conventional reservoirs, sorption-induced swelling results in a large shift in the reservoir's stress, excluding the poroelastic effect. Figure 2 displays the variations between induced stress and radial displacement whether sorption is considered or not,

emphasizing the significance of doing so. It should be noted that the steady reservoir pressure distribution results in negative (tensile) radial and hoop stresses, when only the poroelastic effect is considered, while compressive radial and hoop stresses are induced, including the sorption effect as shown in Figure 2a. The swelling/volumetric strains induced by the sorption of $CO_2$ and $CH_4$ are different, and then the displacement of $CH_4$ by $CO_2$ causes tremendous net swelling. However, the zero displacement at the outer boundary condition overwhelmingly restricts the coal to accommodate the net swelling. In contrast, the induced displacement, considering the sorption effect, is one order more than that merely considering the poroelastic effect in Figure 2b. Thus, considering the strong swelling and boundary condition, the stress state in the whole domain is markedly elevated in Figure 2a. Furthermore, the maximum radial and hoop stresses occur at the borehole; thus, reservoir instability can be analyzed using the wellbore stability issue. Additionally, the incremental reservoir pressure distribution $\Delta p$ in Equation (6), rather than the real reservoir pressure distribution, causes the stresses change.

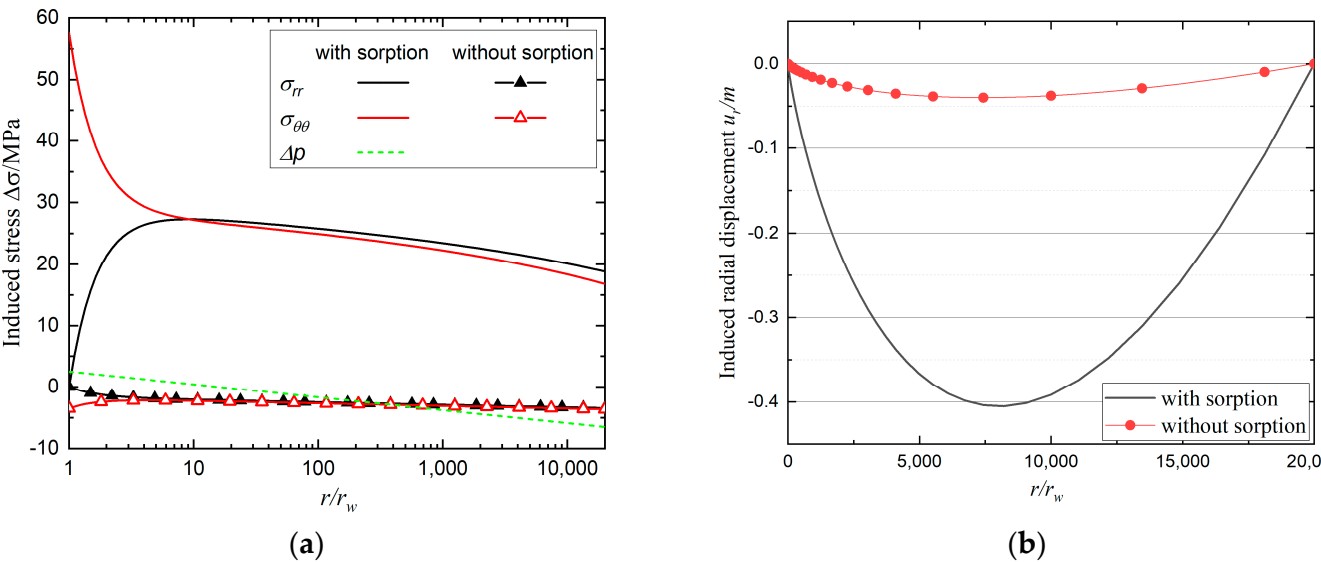

**Figure 2.** (**a**) The induced stress and (**b**) radial displacement difference with normalized radial distance considering sorption or not. In this case, pure $CO_2$ is injected by $p_w$ = 12 MPa with reservoir pressure before injection $p_{res}$ = 3 MPa and ZDBC condition is imposed.

From Equations (12)–(15), with the exception of the reservoir pressure distribution ($p_0$, $p_i$, $p_w$), the induced stress is clearly controlled by the mechanical properties ($E$, $\nu$), adsorption parameters ($P_\varepsilon$, $\varepsilon_L$), and gas component of $CO_2$ ($c_{CO_2}$). Then, the parametric analysis of corresponding factors' effect on the induced stress is made as follows.

### 3.2.1. The Influence of Langmuir-Type Constants ($P_\varepsilon$, $\varepsilon_L$)

The influence of adsorption parameters on the distribution of the radial, hoop and effective horizontal stress variation $\sigma_{rr}^2$, $\sigma_{\theta\theta}^2$, $\Delta\sigma^{eff}$ with normalized radial distance in pure $CO_2$ injection is displayed in Figure 3. Both adsorption parameters $P_\varepsilon$, $\varepsilon_L$ severely affect the stress distribution and subsequent wellbore instability. Additionally, we may deduce that the more $\varepsilon_{LCO_2}$ and less $P_{\varepsilon CO_2}$, the higher the radial, hoop, and effective horizontal stresses can be generated. While the result differs from that under common uni-axial strain condition [15], the induced effective horizontal stress under ZDBC condition varies little throughout the radial distance in some circumstances.

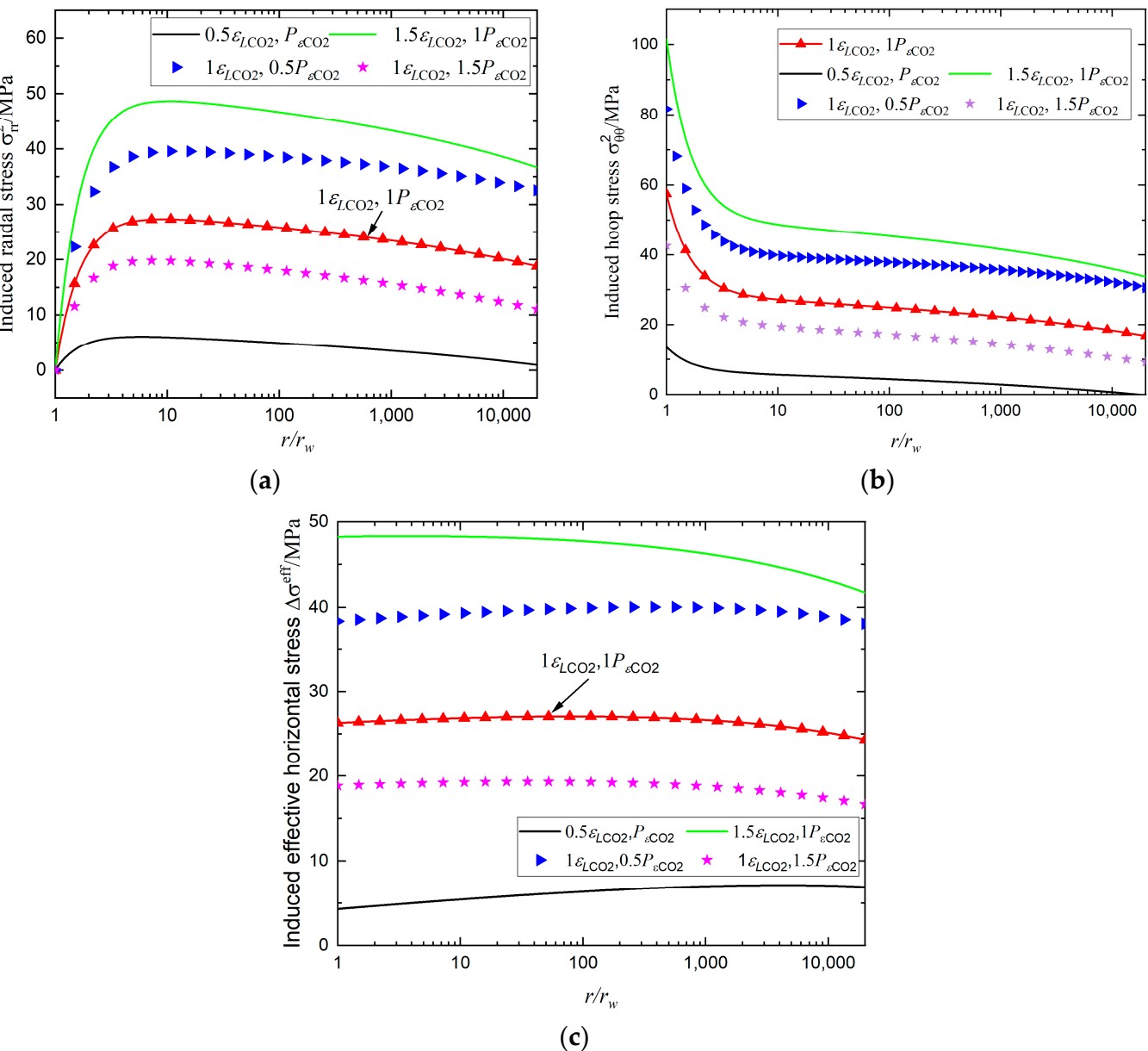

**Figure 3.** Profiles of (**a**) induced radial stress, (**b**) hoop stress and (**c**) effective horizontal stress distribution with varying adsorption parameters. In this case, pure $CO_2$ is injected by $p_w$ = 12 MPa with $p_{res}$ = 3 MPa and ZDBC condition is imposed.

### 3.2.2. The Influence of Young Modulus and Poisson Ratio of Coal

The influence of mechanical properties on the induced radial, hoop and effective horizontal stress distribution is depicted in Figure 4. The maximum values of induced radial, hoop and effective horizontal stresses grow as the Young modulus and Poisson ratio increase. In addition, the Poisson ratio has less substantial effect on the stress distribution than the Young modulus, as shown in Figure 4. Consequently, sorption-induced stress should be given more consideration in coal seams and shale with a high Young modulus and Poisson ratio.

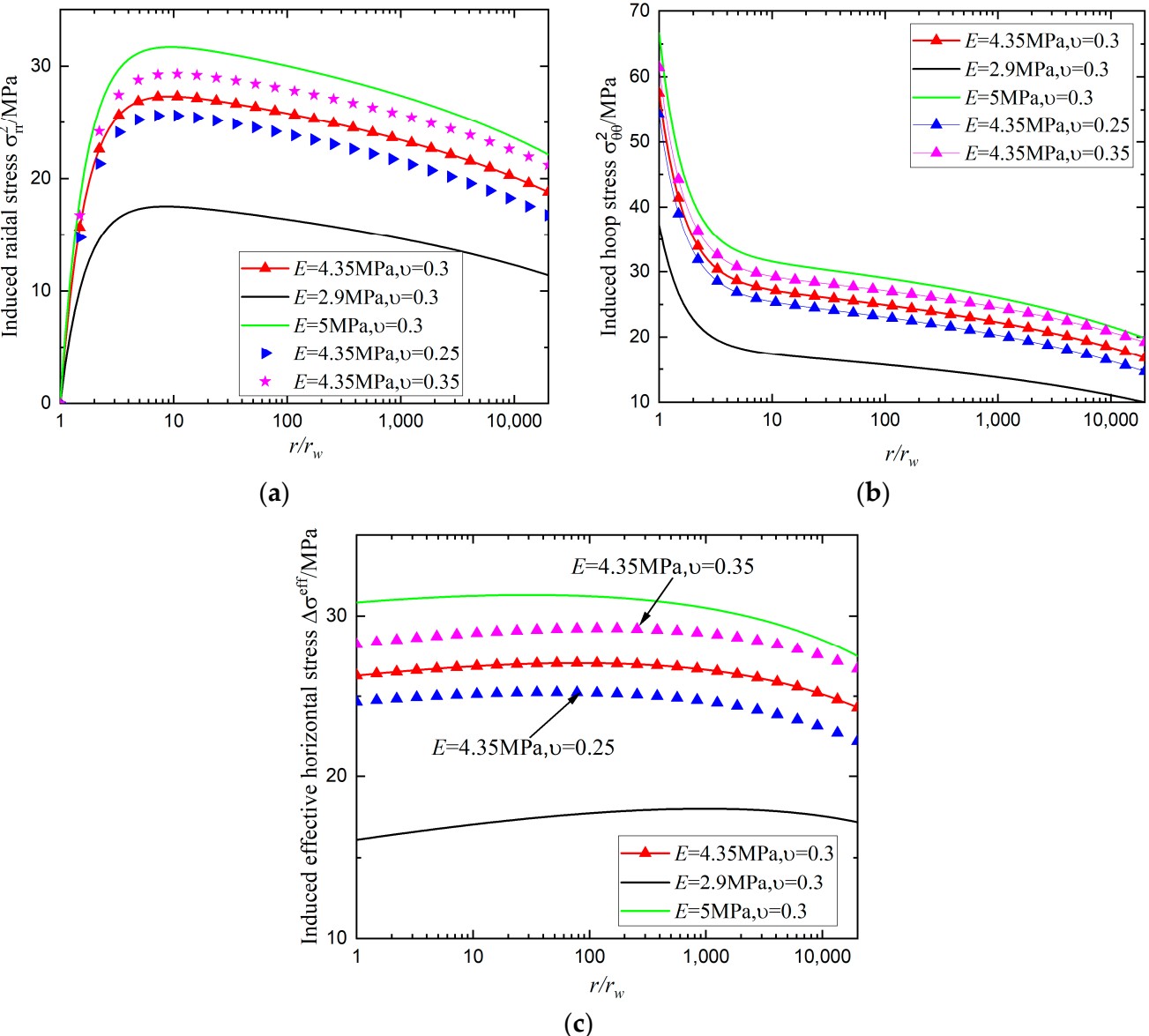

**Figure 4.** Profiles of induced (**a**) radial stress, (**b**) hoop stress and (**c**) effective horizontal stress distribution with varying Young modulus and Poisson ratio. In this case, pure $CO_2$ is injected by $p_w$ = 12 MPa with $p_{res} = = 3$ MPa and ZDBC condition is imposed.

### 3.2.3. The Influence of Gas Component

The coal seams' permeability significantly decreases when pure $CO_2$ is injected, hence mixture gas ($CO_2/N_2$) injection is recommended, while wellbore stability during mixture gas injection has not been fully researched. Due to the distinctive sorption-induced swelling characteristics of $N_2$, $CH_4$, and $CO_2$, mixture gas injection with varied gas components can produce various stress changes and wellbore failure indexes. Figure 5 depicts the induced radial, hoop, and effective horizontal stresses with varied gas components. Since more volumetric fraction of $N_2$ with weakest adsorptive property is injected into the coal seams as $c_{CO_2}$ decreases, maximum values of induced radial, hoop, and effective horizontal stresses become lower. Furthermore, effective horizontal stress near the wellbore becomes negative with $c_{CO_2} = 0.2$, suggesting that the displacement of $CH_4$ with mixture gas ($c_{CO_2} = 0.2$) results in net coal shrinkage in this case.

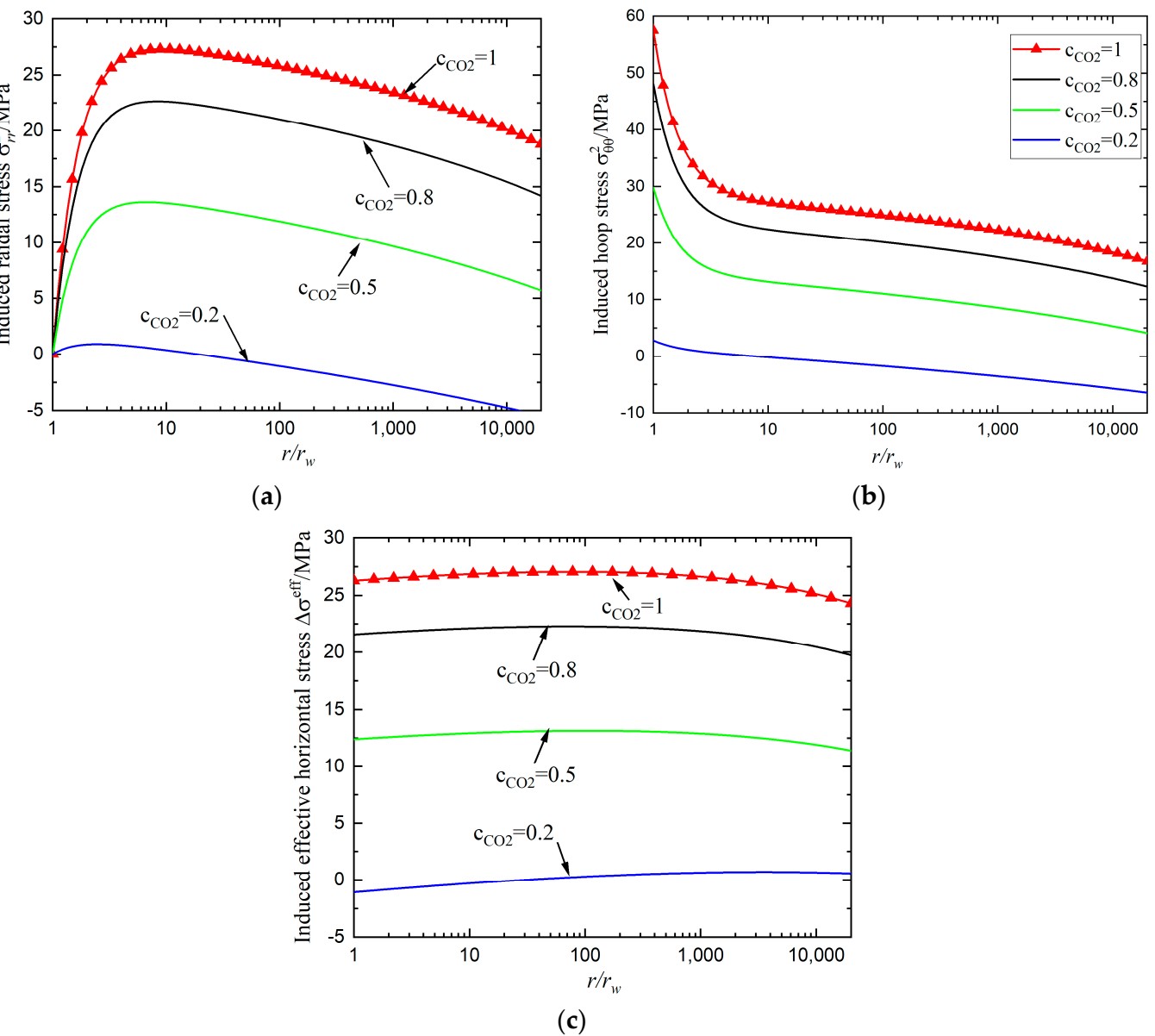

**Figure 5.** Profiles of induced (**a**) radial stress, (**b**) hoop stress and (**c**) effective horizontal stress distribution with varying gas components. In this case, mixture gas is injected by $p_w$ = 12 MPa with $p_{res}$ = 3 MPa and ZDBC condition is imposed.

## 4. Wellbore Stability Analysis during Mixture Gas Injection

The developed methodology is applied to investigate the total stress field around the wellbore and coal failure with the different borehole pressures and gas components of $CO_2$ during $CO_2$-ECBM.

### 4.1. The Total Stress and Failure Index around the Borehole

The preceding investigation indicated that sorption-induced swelling has a significant impact on the stress field during mixture gas injection. In this section, the total stress distribution along with varying gas components $c_{CO_2}$ is examined. Comparisons of Figure 6 revealed that the gas component of $CO_2$ significantly affects the total radial and hoop stress distribution along $0°$ and $90°$ directions, whose difference is caused by initial anisotropic in-situ horizontal stress. Additionally, when the gas component of $CO_2$ $c_{CO_2}$ diminishes, a larger proportion of less adsorptive $N_2$ is absorbed into the reservoir, which lowers radial stress inside the borehole and hoop stress at the borehole. In this instance, sorption-induced

strain turns into net shrinkage in comparison to the initial $CH_4$ adsorption, resulting in a reduction in sorption-induced stresses. Moreover, tensile hoop stress is generated when pure $N_2$ ($c_{CO_2} = 0, c_{N_2} = 1$) is injected into the reservoir, which leads to tensile failure near the wellbore. Similar to but different from thermal fracturing caused by temperature change [30], tensile failure is induced by the displacement of the initial $CH_4$ with weakly adsorptive $N_2$. Tensile failure, which was rarely noticed, should be carefully examined in the mixture gas injection with a large proportion of $N_2$ (including pure $N_2$) into the coal seams.

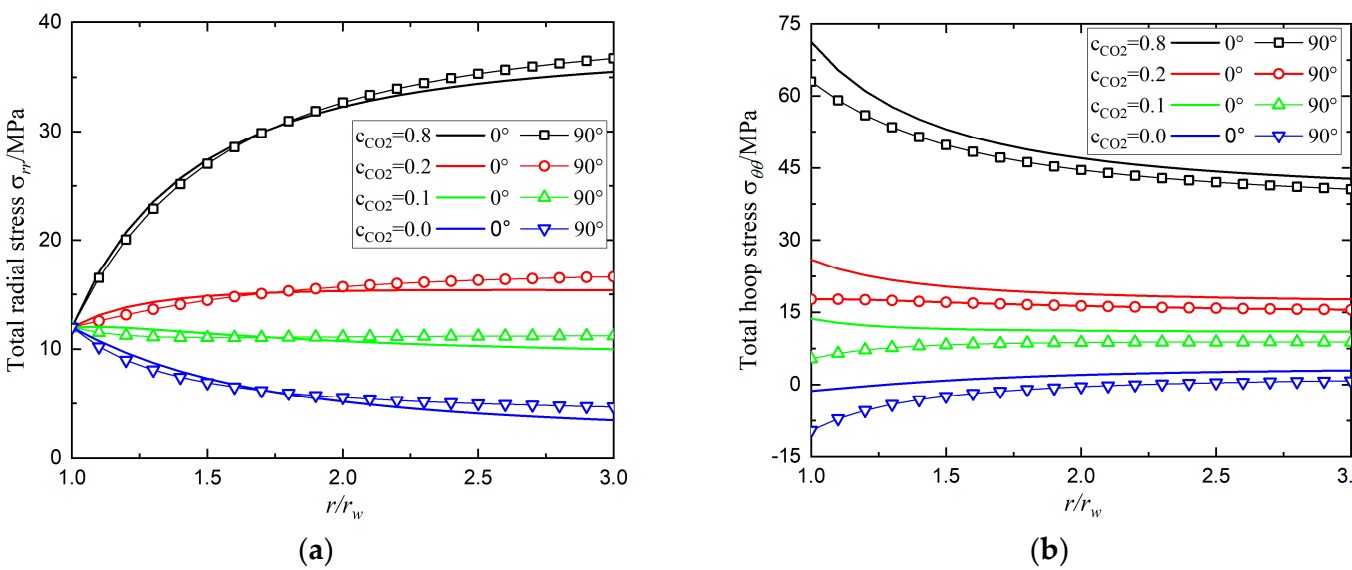

**Figure 6.** Profiles of (**a**) total radial stress and (**b**) total hoop stress with radius with $c_{CO_2}$ = 0.8,0.2,0.1,0 along 0° and 90° direction. In this case, mixture gas is injected by $p_w$ = 12 MPa with $p_{res}$ = 3 MPa and ZDBC condition is imposed.

Next, failure index around the borehole in Equation (23) with various gas components of $CO_2$ is presented in Figure 7. Given the large gas component of $CO_2$ ($c_{CO_2} \geq 0.8$), wellbore failure due to significant sorption-induced stress (seen in Figure 6) emerges at the borehole wall. The maximum and minimum principal stresses at the borehole wall in this case are the hoop and radial stresses (i.e., $\sigma_{\theta\theta} > \sigma_{zz} > \sigma_{rr}$), respectively. Furthermore, since the magnitude of isotropic stress induced by sorption is substantially greater than that of initial in-situ stress, the failure index distribution exhibits negligible anisotropy. Additionally, when $c_{CO_2} = 0.2$, the failure index around the wellbore lies in the range of $-3.1 \sim -6.7$ MPa, indicating a stable wellbore state under the given conditions. Additionally, the highest failure index $F_{max}$ is located at the borehole wall in a 0° direction. In contrast, when $c_{CO_2}$ drops further ($c_{CO_2} = 0.1$), wellbore failure occurs at the borehole wall in the 90° direction. In this instance, a large proportion of weakly adsorptive $N_2$ induces the smallest hoop stress at the borehole in a 90° direction, as shown in Figure 6b, which is the minimum principal stress (i.e., $\sigma_{zz} > \sigma_{rr} > \sigma_{\theta\theta}$). So the wellbore failure index first declines and then climbs as $c_{CO_2}$ goes from 1 to 0.1. Then, we may deduce that for the provided parameters in this situation, the gas component of $CO_2$ must fall within a rational scope in order to preserve wellbore stability.

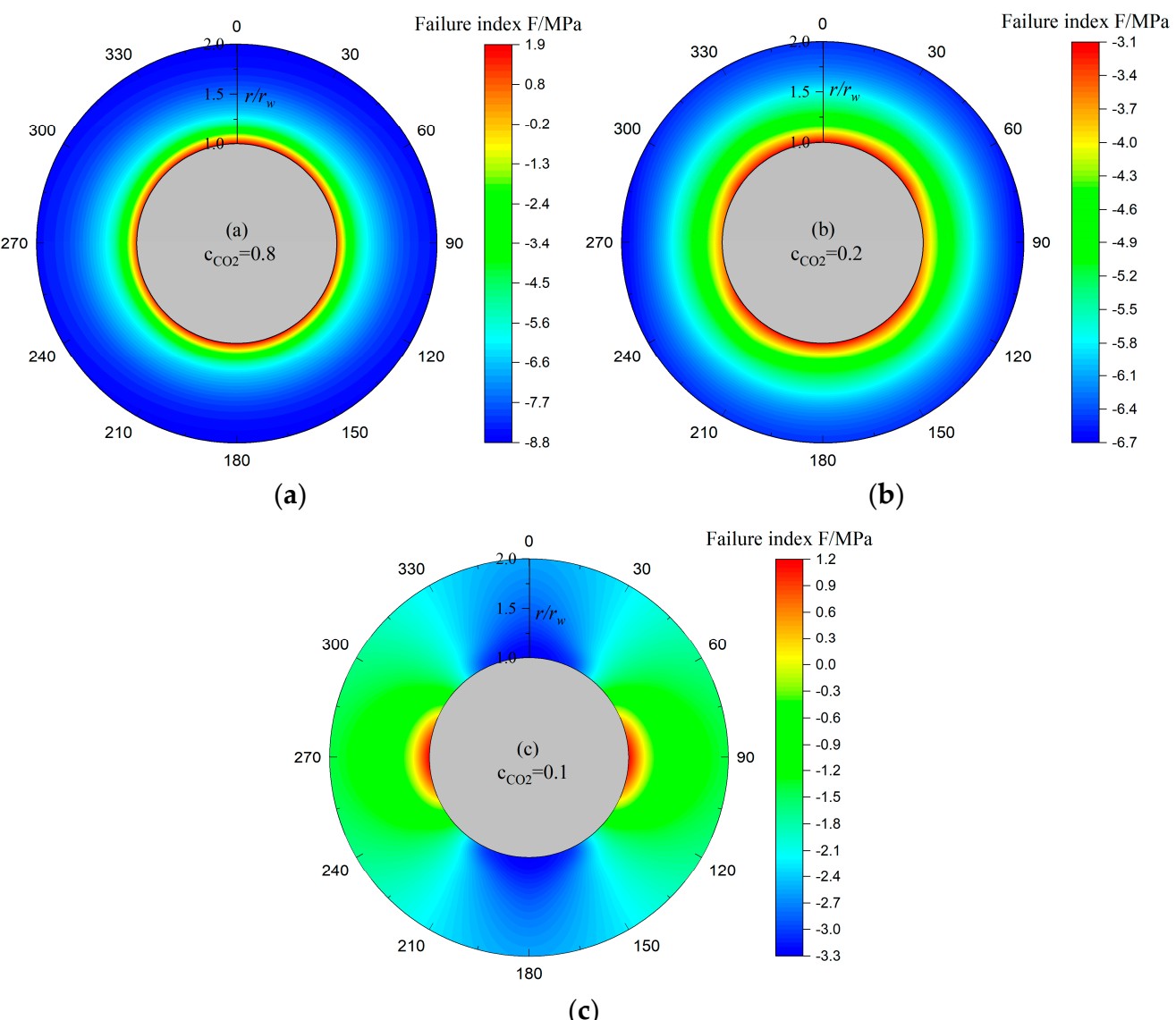

**Figure 7.** Failure index distribution in the vicinity of the borehole with different gas component: (**a**) $c_{CO_2} = 0.8$; (**b**) $c_{CO_2} = 0.2$; (**c**) $c_{CO_2} = 0.1$. In this case, mixture gas is injected by $p_w = 12$ MPa with $p_{res} = 3$ MPa and ZDBC condition is imposed.

Figure 8 depicts the evolution of the maximum failure index with borehole pressure $p_w$ under the varying gas components of $CO_2$ $c_{CO_2}$ to investigate wellbore stability in the mixture gas injection. It should be mentioned that the maximum borehole pressure $p_w$ is constrained to the initial minimum horizontal stress $\sigma_{h0}$ to prevent the reservoir from hydraulic fracturing. In the absence of sorption, as borehole pressure $p_w$ rises, the maximum failure index first sightly decreases and subsequently increases, as illustrated by the violet dashed line in Figure 8. The increase in pore pressure causes effective horizontal and vertical stresses to decline, which leads Mohr's circle of stress to continuously shift to the left and approach the failure envelope, leading to an increase in wellbore failure index. In addition, the evolution of the maximum failure index is irrelevant with gas component $c_{CO_2}$, when just the poroelastic effect is taken into account.

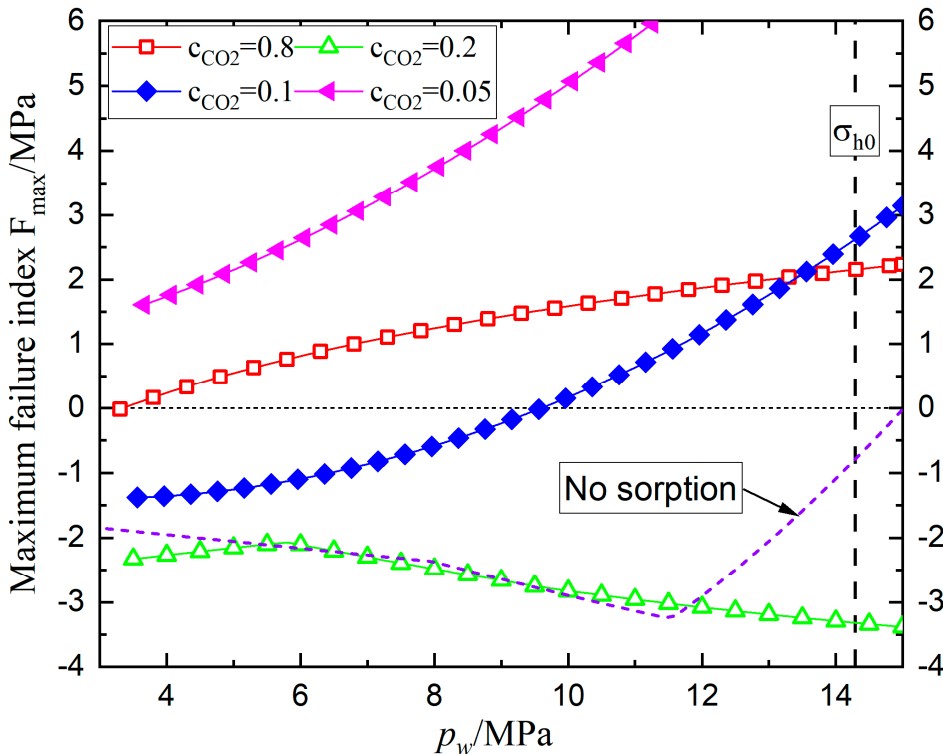

**Figure 8.** Profile of maximum failure index's evolution with varying borehole pressure and gas component. In this case, $p_{res}$ = 3 MPa.

However, the maximum failure index's evolution with mixture gas injection considering sorption-induced swelling differs from that when only the poroelastic effect is considered. Increased borehole pressure causes $F_{max}$ to decline when the gas component of $CO_2$ reduces from 0.8 to 0.2, indicating that the wellbore turns to be stable in the mixture gas injection. Additionally, for the given specific conditions, when $c_{CO_2}$ is higher than 0.8, mixture gas injection induces wellbore instability with arbitrary borehole pressure (certainly higher than the reservoir pressure before injection $p_i$).

In contrast, $F_{max}$ increases as borehole pressure $p_w$ rises when the gas component of $CO_2$ $c_{CO_2}$ continuously decreases form 0.2 to 0.05. As seen in Figure 6b, net shrinkage and correspondingly lower hoop stress are caused by a considerably smaller fraction of $CO_2$ associated with a higher proportion of $N_2$. Thus, the anisotropy of the principal stress state ($\sigma_{zz} > \sigma_{rr} > \sigma_{\theta\theta}$) continues to be magnified, and the related failure index increases, as shown in Figure 8. Furthermore, it should be noted that a large proportion of $N_2$ ($c_{CO_2} = 0, c_{N_2} = 1$) causes tensile hoop stress in Figure 6b, which also sets a lower limit of $c_{CO_2}$ for preventing wellbore tensile failure.

### 4.2. Critical Borehole Pressure and Gas Component (CBPGC)

A workflow with an iterative loop was developed to obtain the critical borehole pressure and gas component of $CO_2$ (CBPGC), maintaining wellbore stability during $CO_2$-ECBM. CBPGC with pore pressure before injection $p_{res}$ = 3 MPa is displayed in Figure 9. The upper limit of $p_w$ is constrained by prohibiting hydraulic fracturing in the coal seams, which is approximately equal to $p_{w\max} = \sigma_{h0}$. Meanwhile, the lower limit of $p_w$ is constrained by $p_w > p_{res}$. Additionally, the wellbore failure index $F_{max} = 0$ restricts the upper and lower limits of the gas component of $CO_2$.

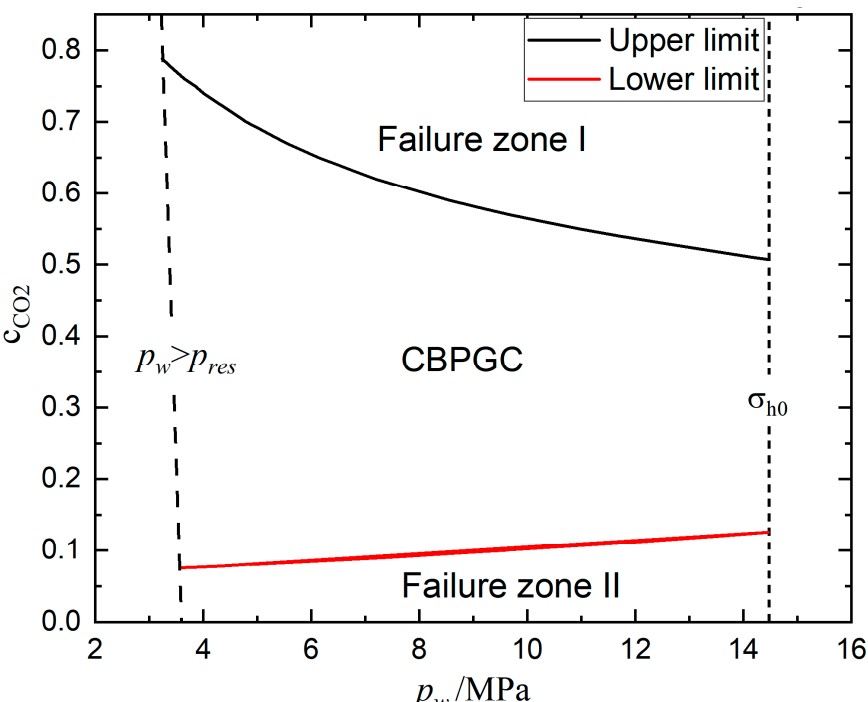

**Figure 9.** CBPGC with pore pressure before injection $p_{res}$ = 3 MPa. The critical $p_w$ in the CBPGC is confined by $\sigma_{h0}$ and $p_w > p_{res}$, and the upper and lower limit of $c_{CO_2}$ in the CBPGC is obtained by wellbore failure index $F_{max} = 0$.

Additionally, wellbore state moves into Failure Zone I in Figure 9, if a mixture gas with a large proportion of $CO_2$ that exceeds the upper limit is injected. In this instance, as shown in Figure 6b, the displacement of initial $CH_4$ with strongly adsorptive $CO_2$ results in considerable compressive hoop stress, and then wellbore failure occurs near the borehole. By contrast, the wellbore state enters Failure Zone II if a mixture gas with a large proportion of $N_2$ (the corresponding gas component of $CO_2$ lower than the lower limit) is injected. Wellbore shear or tensile failure occurs when the initial $CH_4$ is mainly displaced by weakly adsorptive $N_2$, which induces lower compressive and sometimes even negative hoop stress. Therefore, the borehole pressure and gas component of $CO_2$ should be set within the CBPGC to prevent wellbore instability. Finally, the CBPGC can provide a benchmark for in-situ coal seams' $CO_2$ storage capacity in terms of wellbore stability.

## 5. Conclusions and Suggestions

Assuming steady reservoir pressure distribution and ZDBC (zero displacement at the outer boundary) condition, the stress field taking into account the sorption and poroelastic effect is derived in the coal seams with anisotropic in-situ stress state. Then, the critical borehole pressure and gas component (CBPGC) maintaining wellbore stability is obtained by combining with shear and tensile failure criteria. Due to the distinct sorption characteristics of $CH_4$, $CO_2$, and $N_2$, mixture gas injection with variable gas composition results in diverse stress change, as opposed to the case when only the poroelastic effect is considered. The following findings can be obtained from this study:

(1) The stress field is significantly influenced by the boundary condition and sorption-induced swelling characteristics. The ZDBC condition results in larger stress change in comparison to the constant stress condition at the outer boundary. Furthermore, the sorption-induced swelling of pure $CO_2$ relative to $CH_4$ induces compressive radial and hoop stresses, whereas tensile radial and hoop stresses are caused when only the poroelastic effect is considered;

(2) With the exception of the reservoir pressure distribution, mechanical properties and adsorption parameters both influence sorption-induced stresses. The larger sorption-

induced stresses are caused by an increase in the Young modulus and Poisson ratio, which suggests that this effect should be taken into account more in the coal and shale with the high Young modulus and Poisson ratio. In addition, adsorption parameters also directly influence sorption-induced stress. More considerable sorption-induced stresses are induced by the larger $\varepsilon_{LCO_2}$ and smaller $P_{\varepsilon CO_2}$;

(3) The borehole pressure and gas component of $CO_2$ should be restrained by the CBPGC to prevent wellbore from shear and tensile failure. Mixture gas injection with a large proportion of $CO_2$ would result in considerable sorption-induced hoop stress and wellbore shear failure. By contrast, when mixture gas with a small proportion of $CO_2$ is injected, the displacement of the initial $CH_4$ with weakly adsorptive $N_2$ would induce less compressive and even tensile hoop stress. In this instance, wellbore shear or tensile failure occurs, which is rarely noticed. Therefore, in light of wellbore stability, the CBPGC can provide a benchmark for in-situ coal seams' $CO_2$ storage capacity;

This paper derives a semi-analytical solution of stress field and failure index assuming the steady reservoir pressure distribution that occurs after the unsteady seepage stage. So the CBPGC can be treated as an ultimate result of mixture gas injection, and gives the upper and lower limits of $CO_2$ storage capacity bound by wellbore stability. In addition, the injection scheme design should pay more attention to wellbore stability during unsteady seepage stage. Moreover, as cased, cemented wellbore are frequently constructed to maintain wellbore stability/integrity; interface failure and zone isolation should be thoroughly studied in the future.

**Author Contributions:** Conceptualization, H.X. and W.L.; methodology, W.L.; validation, Z.W. and S.Y.; formal analysis and investigation, Z.W.; resources, S.Y. and P.T.; writing—original draft preparation, H.X.; writing—review and editing, W.L.; supervision, W.L.; funding acquisition, W.L. All authors have read and agreed to the published version of the manuscript.

**Funding:** This research was financially supported by the Fundamental Research Program of Shanxi Province, China (Grant No. 20210302124664, 202103021224059), CNPC Science and Technology Project (Grant No. 2021DQ03-37).

**Institutional Review Board Statement:** Not applicable.

**Informed Consent Statement:** Not applicable.

**Data Availability Statement:** Data are contained in the article.

**Conflicts of Interest:** The authors declare no conflict of interest.

## Nomenclature

| | |
|---|---|
| $i$ | Gas type |
| $\varepsilon_v, \varepsilon_b$ | Volumetric strain due to adsorption and deformation |
| $p_{\varepsilon i}, \varepsilon_{Li}$ | Langmuir-type swelling constants |
| $V_{Li}, P_{Li}$ | Langmuir-type adsorption constants |
| $E, \nu$ | Young modulus and Poisson ratio |
| $\delta_{ij}$ | Kronecker's delta |
| $\varsigma$ | Biot coefficient |
| $p_0, p_{res}$ | Initial and depleted reservoir pressure, respectively |
| $p_w$ | Borehole pressure |
| $r, \theta, z$ | Directional index in cylindrical coordinates |
| $r_w, r_b$ | Radius of wellbore and outer boundary |
| $\sigma_{H0}, \sigma_{h0}, \sigma_{V0}$ | Initial maximum, minimum horizontal, and vertical stress, respectively |
| $\sigma_{rr}^1, \sigma_{\theta\theta}^1, \sigma_{zz}^1, \sigma_{r\theta}^1, \sigma_{rz}^1, \sigma_{\theta z}^1$ | Normal and shear stress components due to in-situ stress and borehole pressure |
| $\sigma_{rr}^2, \sigma_{\theta\theta}^2$ | Radial and hoop stress due to poroelastic response and gas adsorption, respectively |
| $\sigma_{rr}, \sigma_{\theta\theta}, \sigma_{r\theta}, \sigma_{zz}$ | Total stress components, respectively |
| $\sigma_{rr}^w, \sigma_{\theta\theta}^w, \sigma_{zz}^w, \sigma_{r\theta}^w$ | Total stress components at the wellbore |
| $\Delta p, \Delta\varepsilon_V$ | Incremental reservoir pressure and volumetric strain, respectively |

| $C_1, C_2$ | Integration constants |
|---|---|
| $F_p, F_\varepsilon$ | Integration constants related to poroelastic response and sorption effect, respectively |
| $F$ | Wellbore failure index |
| $C, \varphi$ | Rock cohesion and friction angle |
| $S_1, S_2, S_3$ | Maximum, median, and minimum stress, respectively |
| $\tau_{oct}$ | Octahedral shear stress |
| $a, b$ | Mogi–Coulomb coefficients |
| $c_{CO_2}, c_{N_2}$ | Gas component of $CO_2$ and $N_2$, respectively |
| CBPGC | Critical borehole pressure and gas component of $CO_2$ |
| ZDBC | Zero displacement condition at outer boundary |
| ZSBC | Constant stress condition at outer boundary |

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
