# Peer review of "Critical Conditions for Wellbore Failure during CO2-ECBM Considering Sorption Stress"

_sustainability, doi:10.3390/su15043696_

Round 1
Reviewer 1 Report
The authors present a semi-analytical method to investigate the wellbore failure index during reservoir-scale CO2 enhanced coalbed methane. The topic is interesting, and the document is well organized. I will suggest a minor reversion decision. A few comments are given for your review:
1. Please give the full name of ECBM when it first appears.
2. Section 2 title can be further briefed.
3. Please add a brief description before you give assumptions in section 2.2.1. Also, add some introduction sentences between the gap between the section and subsection.
4. A description is needed to explain the influence of assumptions taken to develop a semi-analytical method.
5. Table format should be addressed.
6. What happened at the connection of page 7 and page 8? It looks like something is missing.
7. Why does the induced stress in figure 2 change dramatically with sorption?
8. Do you validate your sorption model with experimental data?
9. The position of the legend should be adjusted to avoid obscuring the data.
Reviewer 2 Report
This paper developed a semi-analytical solution of coal wellbore failure index during mixture gas (CO2/N2) injection, which considered the poroelastic response and sorption effect. sorption stress have significant effect on the stress field due to pore pressure change and adsorptive swelling, which is not well studied. This is a meaningful work and helpful to accurately evaluate the critical injection pressure and gas component constrained by the wellbore stability. The manuscript is recommended to be minor revision, and the following modifications need to be made:
(1) Improve the language and unify the nomenclature in the paper, such as “borehole pressure” , “wellbore pressure” and “injection pressure”.
(2) Add the list of symbols.
(3) “the safety of CO2-ECBM” in the abstract may be revised as “ the safety and sustainability of the CO2-ECBM process”, since an undamaged borehole is fundamental to the prolonged injection.
Reviewer 3 Report
In this paper, a semi-analytical model of coal wellbore stability during CO2-ECBM process is established to predict the critical borehole pressure and gas component preventing the wellbore stability, which is of great importance to assure the safety of CO2-ECBM process. The model incorporates the impacts of mixture gas (CO2/N2) adsorption and poroelastic response due to the mixture gas injection, and different boundary conditions including zero displacement and constant stress are investigated. This work is valuable and pioneering, and can give a quantitative evaluation of CO2 storage capacity from the perspective of wellbore stability. Therefore, the manuscript is recommended to be minor revised, and my detailed comments are given as follows:
(1) Add the list of symbols to the paper.
(2) Use the same professional terms, such as “pore/reservoir pressure” and “injection/borehole/wellbore pressure”.
(3) The sub-sub sections “2.2.2” and “2.2.3” can add the corresponding “Mode 1/2” to increase the readability.
(4) The number 15 in line 285 may denote a reference, and revise the format.
